Assessing the utility of urinary and fecal cortisol as an indicator of stress in golden snub-nosed monkeys (Rhinopithecus roxellana)

Chen Haochun 1 2
Yao Hui 3
Yang Wanji 1
Fan Penglai 1
Xiang Zuofu zorph@163.com 1 2
1 Institute of Evolutionary Ecology and Conservation Biology, Central South University of Forestry & Technology , Changsha , Hunan , China
2 College of Life Science and Technology, Central South University of Forestry & Technology , Changsha , Hunan , China
3 Key Lab of Conservation Biology for Shennongjia Golden Snub-nosed Monkeys, Hubei Province , Shennongjia Forest District , China
Hopper Lydia
Electronic publication date: 2017 Aug 21
Publication date: 2017
Volume: 5
Electronic Location ID: e3648
Received 2016 Oct 24; Accepted 2017 Jul 13
Copyright: ©2017 Chen et al.
Copyright year: 2017
Copyright holder: Chen et al.
License: This is an open access article distributed under the terms of the Creative Commons Attribution License, which permits unrestricted use, distribution, reproduction and adaptation in any medium and for any purpose provided that it is properly attributed. For attribution, the original author(s), title, publication source (PeerJ) and either DOI or URL of the article must be cited.
License URL: https://creativecommons.org/licenses/by/4.0/

Keywords: Stress, Cortisol concentration, Lag time, Golden snub-nosed monkeys

Funding: National Natural Science Foundation of China 31670397 National Key Technology R & D Program of China 2013BAD03B03 2016YFC0503200 This study was supported by the National Natural Science Foundation of China (31670397), and the National Key Technology R & D Program of China (2013BAD03B03, 2016YFC0503200). The funders had no role in study design, data collection and analysis, decision to publish, or preparation of the manuscript.

==============================
Cortisol concentration (CC) is often used as a stress indicator in animals, as high CC is associated with elevated stress levels. During field research, non-invasive methods of measuring CC, such as collection of urine and feces, are superior to using blood samples when monitoring free-ranging animals’ stress levels. However, due to different metabolic pathways, whether CC can be detected in urine and feces to reliably assess stress varies across species. Therefore, it is important to ascertain whether urine and fecal samples are a reliable source for determining CCs and to determine a suitable sampling regime. In this study, we subjected three captive adult golden snub-nosed monkeys (Rhinopithecus roxellana) to a high-stress situation (capture and injection). Urine and feces were collected for four days before and for four days after the manipulations for laboratory analysis. Immunoreactive CC was detected with a commercial enzyme immunoassay (EIA) kit and showed distinct rises. Peak CC values in urine were detected within 5 h, while peak fecal CC ranged between 5 and 24 hours post-interference. These results provide evidence that CC in urine and feces can be used to assess stress levels in the golden snub-nosed monkey. The optimal time frame to collect urinary and fecal samples for CC analysis is within one day of a potential stressful event.

Introduction

Cortisol, the primary glucocorticoid of primates, is released in response to stress (Fagot et al., 2014; Whitten, Brockman & Stavisky, 1998). Measuring cortisol levels in blood (serum/plasma) has proved to be a useful indicator of perceived stress (Broom & Johnson, 1993). Nonetheless, collecting blood from wild animals is not easy, involving stressful procedures like restraint and sedation. Hence, non-invasive methods such as collecting urine and feces to assess the stress response are becoming prevalent (Behie, Pavelka & Chapman, 2010; Möstl & Palme, 2002; Novak et al., 2013; Whitten, Brockman & Stavisky, 1998). Although cortisol is the prominent biological active glucocorticoid in most primate species, only traces of native cortisol may exist in the urine and feces of particular species, as it is usually converted into various metabolites before being excreted (Hämäläinen et al., 2014; Möstl & Palme, 2002). For instance, Bahr et al. (2000) measured cortisol and several metabolites, and found that native cortisol was a major urinary excretory product in common marmosets (Callithrix jacchus), while only small amounts were present in the urine of long-tailed macaques (Macaca fascicularis) and chimpanzees (Pan troglodytes). Conversely, fecal cortisol could barely be detected in common marmosets (Bahr et al., 2000). As a consequence, a respective assay must be validated for any particular species and matrix to ensure proper quantification of cortisol or its metabolites (Sheriff et al., 2011).

Meanwhile, knowing the lag time between the secretion and excretion of glucocorticoids is beneficial to better understand the connection between physiological alterations and behavioral events/states, but the lag time is also matrix dependent and variable across species. In urine, the lag time is comparatively consistent for primates; 2.5 h in common marmosets, 4.8 h in chimpanzees, 5.5 h in long-tailed macaques, and 4.5 h in baboons (Papio cynocephalus cynocephalus) (Bahr et al., 2000; Wasser et al., 1994). In contrast, the lag time for the appearance of cortisol in feces is both longer and more variable among species, since defecation frequencies are affected by diet and gut passage time. Although not always the case (Goymann, 2012; Hämäläinen et al., 2014), hormones are more slowly excreted in the feces of larger-bodied primate species. Hormones excreted in feces showed a delay of 44 h in Western lowland gorillas (Gorilla gorilla gorilla) (Shutt, Setchell & Heistermann, 2012), 26 h in olive baboons (P. cynocephalus), 22 h in chimpanzees (Wasser et al., 2000), and 8–16 h in squirrel monkeys (Saimiri sciureus) (Moorman et al., 2002). Thus, for each species, it is also important to identify the lag time for cortisol excretion (or its metabolites) besides validating the method to monitor the stress response.

The golden snub-nosed monkey (Rhinopithecus roxellana) is an Asian colobine endemic to China. Its discontinuous geographic distribution occurs in the provinces of Sichuan, Gansu, Shaanxi, and Hubei (Li, Pan & Oxnard, 2002). It is listed as Endangered by the International Union for Conservation of Nature (IUCN, 2014). Golden snub-nosed monkeys live in a multilevel or modular society, in which several one-male and multi-female units with one or several all-male units form a band that feeds, forages, travels and rests together (Zhang et al., 2006). Bachelor males in all-male units have to fight for dominance, while the resident males in one-male units face mating competition and risk of being deposed by the bachelor males. Furthermore, primates living in the wild routinely experience stressful situations including dominance interactions, diseases, parasitism, predation, and food shortages (Novak et al., 2013). With the development of ecotourism, golden snub-nosed monkeys may also experience stress from close proximity to humans (Maréchal et al., 2011; Xiang et al., 2011). Hence, the possibility to monitor the physiological state of golden snub-nosed monkeys would be valuable option to evaluate their health and well-being. However, no published studies have validated whether urinary or fecal cortisol or its metabolites can be used as indicators of perceived stress in R. roxellana. In this experiment, by handling three golden snub-nosed monkeys and injecting a saline solution we stimulated a potential stress response, which was subsequently measured by enzyme immunoassay (EIA). Our aims were to validate an EIA for monitoring urinary and fecal cortisol concentrations (CC), and to determine the lag time of urinary and fecal CC for captive golden snub-nosed monkeys.

Methods

Ethics statement

Prior to conducting this study, approval was gained from the Shennongjia National Nature Reserve (snnr-081201), and the Institutional Animal Care and Use Committee of Central South University of Forestry & Technology (csuft-090120).

Animals and housing

The animals used in this study had been rescued from illness or injury and were being reared in cages, as they were not yet ready for reintroduction into the wild. Three adult golden snub-nosed monkeys, two males (QQ and TT) and one female (SN), were chose as subjects at Xiaolongtan conservation station, Shennongjia National Park (SNP), Hubei, China. QQ was housed with two females and an infant. TT was housed with SN. Of the three, only SN was captive-born at Xiaolongtan. Each enclosure was 25 m2 in area, 5 m high, and contained a dead tree fixed in the middle. There was a small cage connected to each enclosure for resting and sleeping. The enclosures were built 20–30 centimeters above a cement foundation. Rails were 3–6 m away from the enclosures to keep tourists away from the animals. The animals were consistently fed peaches, apples, and other similar foods three times a day throughout the experiment. Water was available ad libitum.

Validation experiment

We imposed a potentially acute stressor (capture and injection) on the three focal animals by entering their cages with several reserve employees, trapping the monkeys with large bags, and thus restraining them before giving each one a saline intramuscular injection. The interventions were conducted in sequence, and we could not stop the monkeys from watching the capture of other monkeys. Each capture took less than 15 min, and the entire procedure was conducted within an hour. Due to the acute stress, the monkeys breathed heavily and defecated once or twice. However, after injection, subjects appeared to recover in minutes, and aside from mild diarrhea in QQ no abnormal behavior was recorded that afternoon. Prior to this intervention, all subjects had been habituated to the presence of investigators and the collection procedure for urine and fecal samples for 20 days, and no investigator took part in the capture to reduce the potential of additional perceived stress during the post-intervention period. The monkeys had never experienced this intervention procedure prior to this study.

Sample collection and storage

Samples were obtained during the day for four days before and four days after the stress manipulation. Two investigators stood alongside each enclosure for sampling from 7:00–12:00 and again from 13:00–18:00. Samples were always collected from outside of the enclosure to avoid potential disturbance of the monkeys; as they usually rested and defecated near the edges of the cage. We collected no more than 4/3 urine/fecal samples for each individual per day, but during the first two days post-injection, we collected every available sample with help of colleagues except for overnight and early morning defecation (before 7: 00), which resulted in a total of 31 urine samples for QQ (13 urine samples pre-intervention, and 18 urine samples post-intervention), 47 for TT (15 urine samples pre-intervention, and 32 urine samples post-intervention), and 30 for SN (16 urine samples pre-intervention, and 14 urine samples post-intervention); as well as 31 fecal samples for QQ (10 fecal samples pre-intervention, and 21 fecal samples post-intervention) and 16 for TT (seven fecal samples pre-intervention, and nine fecal samples post-intervention). The collected number of fecal samples for SN was insufficient (n = 9), especially as no fecal sample of SN was collected in the first 30 h post injection, so we excluded this sample set from all subsequent analyses.

Fresh feces were collected from the clean, dry cage floor using steel clamps and immediately placed in a Ziploc bag. The clamps were washed and dried after every collection. A minimum of 0.5 mL of urine were collected in a disposable plastic bag attached to a long stick (Fig. 1). Holding the stick, we collected the monkeys’ urine in the bag, thereafter transferring the urine into centrifuge tubes by syringe. If the bag was not placed in time to catch falling urine, urine on the floor was collected using a syringe if it was in reach of the investigators. Urine and feces were discarded in the event of cross-contamination with each other or with water. All samples were collected and analyzed separately. Once collected, samples were stored in a portable ice box filled with ice bags until they could be placed in the freezer (−20 °C) within four hours of collection. Samples were kept frozen until hormone analyses were performed at Central South University of Forestry and Technology.

Pre-treatment of samples

Urine samples were centrifuged at 4,000 rpm for 15 min after thawing at ambient temperature. Then the supernatant was diluted with assay buffer 400 times prior to cortisol assay and 20 times prior to creatinine assay, in order to make the results fall within the range of the respective standard curves (dilution ratios were determined previously using pilot assays).

Figure 1 Urine sampling device.

Fecal samples were processed based on the method described by Wasser et al. (2000), and Fan et al. (2013), and the instructions provided by the EIA kit manufacturer. Fully lyophilized, powdered fecal samples (0.1 g) were put into 1.5 mL centrifuge tubes containing 1 mL of ethanol (100%). After 30 min of shaking, samples were centrifuged at 4,000 rpm for 15 min. The supernatant was transferred into a clean tube, then evaporated to dryness in a 60 °C; water bath. Extracted samples were re-dissolved with 100 μL ethanol, followed by 900 μL of Assay Buffer (AB, PBS added bull serum albumin). A volume of 100 μL was taken out and diluted with 200 μL AB prepared for assay.

Enzyme immunoassay (EIA)

Cortisol concentration was assessed with a commercial EIA kit (catalogue #K003-H5) from Arbor Assays (Ann Arbor, MI, USA). According to the manufacturer, cross reactivity of the cortisol antibody is 100% for cortisol, 18.8% for dexamethasone, 7.8% for prednisolone, 1.2% for both corticosterone and cortisone, and less than 0.1% for progesterone. Intra-assay coefficients of variation are 6.5% (n = 5) and 7.8% (n = 5) for high- and low-concentration quality controls. Inter-assay coefficients of variation are 9.3% (n = 5) and 10.2% (n = 5) for high- and low-concentrated quality controls. Assay protocols were based on the product instructions, except that standards were at 3,200, 1,600, 800, 400, 200, and 100 pg/mL in urine cortisol assays, but were halved in assays for fecal samples to get better results. Optical densities were read at 450 nm with a plate reader (DNM 9602, Beijing Pulang New Technology Co., Ltd, Beijing, China). Cortisol levels were calculated using an online four-parameter logistic curve-fitting program the manufacturer provided.

To determine the degree of parallelism for the EIA, a fecal extract pool and a urine pool were serially diluted in AB buffer, assayed, and compared with the respective standard curve. Results were plotted as the percentage bound vs. the log concentration measured.

Assay accuracy was assessed for urine and feces respectively. A urine/feces pool containing low CC was mixed with another urine/feces pool containing high CC (the CC was determined in pilot assays) in different ratios (2:8, 4:6, 6:4, and 8:2) and subsequently analyzed. Regression curves of measured and expected cortisol concentrations are presented in Fig. 2.

Figure 2 Validation results for detecting urinary and fecal cortisol in the golden snub-nosed monkey (Rhinopithecus roxellana).

(A) Parallelism test; the B/B0% were calculated by optical densities of standard samples comparing to optical densities of blanks then multiplied by 100%. (B) Accuracy test.

Determination of creatinine

To adjust for variations in water content, urinary CC was indexed against creatinine and expressed as μg/mg Cr. Creatinine level was determined by a urinary creatinine detection kit from Arbor Assays (catalogue #K002-H5) based on the Jaffe reaction (Taussky, 1954). The optical density was read at 490 nm with a plate reader (DNM 9602, Pulang, Beijing). Creatinine levels were calculated using an online four-parameter logistic curve-fitting program the manufacturer provided.

Statistical analysis

The degree of parallelism of serial dilutions of steroid extracts to the standard curve were assessed with an ANCOVA for testing whether sample pool curves were similar to a standard curve, comparing slopes and intercepts respectively. Assay accuracy was assessed by mean percent recovery. Percent recovery was calculated based on measured CC dividing by expected CC. A mean percent recovery between 90% and 110% was determined to be an acceptable degree of accuracy.

Lag time was defined as the time between the stress manipulation and the occurrence of the highest individual signal of immunoreactive CC post-manipulation. The time at which the injections were conducted was designated as Time = 0. Individual baseline CCs were expressed as median (±the interquartile range). The Mann–Whitney U-test with Bonferonni correction was used to compare individual differences in baseline CCs. We eliminated the urine samples collected the same day after the injection and the fecal samples collected for two days after the injection when calculating the medians and performing the Mann–Whitney test.

Data were processed in Microsoft Excel 2010 and SPSS 19.0. Two-tailed significance levels were set at p = 0.05.

Results

Serial dilutions of pool samples yielded similar curves to the standard cortisol curve (Fig. 2A, for urine: slope, F1,6 = 0.02, p = 0.964; intercept, F1,7 = 1.075, p = 0.334; for feces: slope, F1,6 = 2.073, p = 0.204; intercept, F1,7 = 0.028, p = 0.871). In the accuracy test, the values of r2 were 0.9900 for urine samples and 0.9949 for fecal samples. Mean percent recovery was 100.8% (n = 4) for urine samples and 96.3% for fecal samples (n = 4).

The individual urinary baseline CCs of TT, QQ, and SN were 1.39 (±0.69) μg/mg Cr (n = 39), 0.54 (±0.60) μg/mg Cr (n = 26), and 0.47 (±0.57) μg/mg Cr (n = 29), respectively. The post-stressor peak urinary CCs increased about 10-fold (13.18 μg/mgCr) for TT, 5-fold (2.62 μg/mg Cr) for QQ, and 11-fold (5.03 μg/mg Cr) for SN above their individual baseline levels. The fecal cortisol baseline levels of TT and QQ were 30.95 (±8.20) ng/g feces (n = 11) and 14.87 (±16.23) ng/g feces (n = 22) respectively, and peak fecal CCs increased about 3-fold (80.28 ng/g feces) for TT and 6-fold (86.16 ng/g feces) for QQ. TT had significantly higher baseline urinary and fecal cortisol levels compared to QQ (urine, U39,26 = 74, p < 0.001; feces, U11,22 = 43, p < 0.05), and significantly higher baseline urinary cortisol levels than SN (U39,29 = 120, p < 0.001). No significant difference was found between QQ and SN’s baseline urinary cortisol levels (U26,29 = 352, p > 0.05).

Peak immunoreactive CCs in urine appeared 3.5 h (n = 3, SD = 1.6) post-injection (Fig. 3). Peak immunoreactive CCs in feces were detected at 5 h in TT, and at 22.9 h in QQ (Fig. 3).

Figure 3 Longitudinal profile of urinary and fecal cortisol concentrations for three golden snub-nosed monkeys (Rhinopithecus roxellana) (TT, QQ, and SN) following stress manipulation.

(A, TT; B,QQ; C, SN).

Discussions

Based on the responses of three captive adult golden snub-nosed monkeys in a high-stress situation (capture and injection), our results clearly demonstrate the suitability of the EIA to reliably detect alterations in immunoreactive CC in urine and feces of golden snub-nosed monkeys.

The average lag time for urinary peak CC was 3.5 h (n = 3) for this study, which is consistent with results from earlier studies (Bahr et al., 2000; Smith & French, 1997), usually indicating lag times for a respective signal in urine of 2–6 h. The lag time for fecal cortisol peak CCs in the present study were 5 and 23 h. This result is similar to findings for chimpanzees, olive baboons, and long-tailed macaques, demonstrating a respective lag time for urinary glucocorticoid output between 8 and 26 h (Bahr et al., 2000; Wasser et al., 2000). However, we believe that for golden snub-nosed monkeys the usual lag time for fecal clearance is closer to 23 h than it is to the 5-hour result for three reasons. First, TT, on an individual level, might have been more susceptible to the imposed stressor than QQ or SN, potentially leading an increased metabolic rate or gastro-intestinal motility, resulting in quicker hormone excretion through feces (Goymann, 2012; Steinbrook, 1998). Secondly, we might have missed the actual peak sample if it was voided overnight or in the early hours of day post-intervention. Thirdly, TT had urinary and fecal baseline CCs more than twice as high as those of QQ, and urinary baseline CCs nearly three times higher than the female monkey SN. The comparatively higher baseline CCs of TT indicate that TT might perceive more stress than the other two individuals, although we can’t exclude the possibility of an individual difference in baseline levels. However, both lag time and baseline level of urinary CCs are comparable between QQ and SN, which might indicate neglectable differences in potentially existing sex-related differences in steroid metabolism.

In conclusion, we have validated a reliable EIA method to monitor CC which can be used in future study. Cortisol lag time is 3.5 h in urine, and 23 h in feces, meaning corresponding CC changes to a certain stressor would shortly show in urine, but probably in the next day in feces. Therefore, we recommend the use of a urine sample if it is possible.

Supplemental Information

Data S1 Raw data

The raw data for Figs. 2 and 3.

Click here for additional data file.

We thank the Administration Bureau of Shennongjia National Park for their support in field work. We thank Bo Zhang, XuejunLuo, Hanlong Chen, Ruoshuang Liu for collecting samples.

Additional Information and Declarations

Competing Interests

Author Contributions

Animal Ethics

Data Availability

The authors declare there are no competing interests.

Haochun Chen conceived and designed the experiments, performed the experiments, analyzed the data, wrote the paper, prepared figures and/or tables, reviewed drafts of the paper.

Hui Yao, Wanji Yang and Penglai Fan performed the experiments, reviewed drafts of the paper.

Zuofu Xiang conceived and designed the experiments, analyzed the data, wrote the paper, prepared figures and/or tables, reviewed drafts of the paper.

The following information was supplied relating to ethical approvals (i.e., approving body and any reference numbers):

Approval was gained from the Shennongjia National Nature Reserve (snnr-081201), and the Institutional Animal Care and Use Committee of Central South University of Forestry & Technology (csuft-090120).

The following information was supplied regarding data availability:

The raw data has been supplied as Data S1.

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
