# Peer review of "Assessing the utility of urinary and fecal cortisol as an indicator of stress in golden snub-nosed monkeys (Rhinopithecus roxellana)"

_PeerJ, doi:10.7717/peerj.3648_

## Round 0.1 · original submission · Major Revisions

Reviewer one is positive about the general approach of your study, but raises some methodological issues. Reviewer two questions the validity of the study due to the fact that only one assay was tested on a low sample size and with an extreme stressor. He thus questions the reliability of the used assay for the detection of lower stress responses.I agree with his point of view and this fact needs to be appropriately stressed in the discussion and statements about the usefulness of the assay revised accordingly.

In addition to the different points raised by the two reviewers, I have the following issues:

In general, I would like to see more information on the study species, e.g. social system (i.e. social challenges they might face), habitat they live in (i.e. ecological challenges they might face) as the use of stress hormone measurements is not limited to conservation but is also applied for studies on behavioural ecology, reproductive function etc.

Line 49-51: Baseline levels cannot be extrapolated across all studies on a species as long as recovery rates are not evaluated. So I doubt the usefulness of a general baseline determination.

More information is needed in the method section:

- Important: describe statistical analysis in the method section

- How were the monkeys caught and handled??
- Explain sample collection in more details. Obviously, for urine, a special device was used. Describe the device and use of it in the text. How much urine was collected? Did you merge urine samples? How did you get to the faecal samples? Did someone have to enter the cage for sample collection?

- You seem to have made some adjustments to the methods based on results previous tests (e.g. dilution of samples). Please describe these.

The reference list needs revision for consistency. Sometimes the full journal title is written in capital letters sometimes not and sometimes only abbreviations are given. This reference is incomplete: Hodges K, Brown J, Heistermann M. 2010. Endocrine monitoring of reproduction and stress. Wild mammals in captivity: principles and techniques for zoo management: 447-468.

Reviewer 1 ·

Basic reporting

Please see General Comments to the Author

Experimental design

Please see General Comments to the Author

Validity of the findings

Please see General Comments to the Author

Additional comments

The manuscript by Chen et al. describes the validation of an enzyme-immunoassay for determining urinary and faecal glucocorticoid metabolite concentrations in golden snub-nosed monkey. The general approach is sufficient, but a number of methodological aspects would need further clarification or additional information. To assist the reader, it would also be advisable to request the assistance of a native speaker to improve the language quality of the MS.

Line 1: Can the authors please explain what they mean by ‘effectivity’. Further, the authors have to keep in mind that they did not validate the link between urinary and faecal cortisol and stress, but rather a test system to monitor respective physiological indicators. I suggest complete revision of the title

Line 10: The study did not validate stress – please rephrase

Line 25: The study did not include any analyses via HPLC or GCMS which could have supported the characterisation of respective GC metabolites in golden snub-nose excreta. Thus the authors cannot draw any conclusion regarding the presence of cortisol in urine and faeces of the study species. Please rephrase the statement.

Line 29: Could the authors please specify ‘dramatic’ or alternatively rephrase

Line 30: I suggest adding ‘… hours post-interference’

Line 38: Should read either ‘…of a stress response’ or ‘ …of stress responses’

Line 42-43: Please specify to which species you are referring to.

Line 42ff: The authors should keep in mind that defecation frequency is an equally important factor in this regard. I thus suggest that respective references should also be referred to in terms of numbers of samples produced post-stimulation.

Line 48: I don’t understand the link to baseline GC levels here. Although it is indeed necessary to determine an individual baseline for calculating respective stress-related responses (see below), this relationship has not been introduced at this stage in the Introduction. I suggest re-formulate.
Line 58: I suggest to exchange ‘employed’ with ‘used’

Line 59: Please specify what you mean by ‘…to analyse results…’

Line 79: The information given in the first half sentence appears somewhat out of context – please re-phrase

Line 81-84: Respective information belongs to the Ethics statement (line 64ff) in my opinion

Line 87ff: Please indicate how many urine and faecal samples have been collected in total per individual. Further please indicate how many samples could not be collected or used (line 93-94). Were there any samples voided over night? If so, please indicate number and frequency.

Line 92: Fig1 can be omitted in my opinion.

Line 93: I suggest rephrasing as follows ‘…was collected by using a…’

Line 93: Please specify ‘within reach’ Again mention number of samples not collected as an indicator for potentially missing peak samples post-interference.

Line 94: Please indicate if samples haven been stored immediately on ice. If not, I suggest conducting a so called storage experiment (e.g. Ganswindt et al. 2012, Afr Zool 47: 261-269)

Line 105: If applicable, please indicate any extraction procedures for collected urinary material. If urine was used natively (as assume), please indicate as well. Subsequently please adjust title of the paragraph.

Line 106ff: To assist the reader, please describe the respective procedure for urine and faecal samples separately.

Line 119-121: Accuracy of an assay, as defined by the assay’s ability to detect the correct amount of hormone in the sample, is usually determined by spiking samples with a defined hormone concentration. Please provide a respective reference(s) for the approached used.

Line 127: I can’t find any information on Data analysis. Thus a respective paragraph has to be included at the end of the M&M section. Here the authors should describe in detail (or give respective reference) how the 3 individual data sets haven been statistically analysed including determination of individual GC baseline levels.

Line 130-131: In line with my comment line 119-121 I wonder how the accuracy can be expressed by comparing slopes, as it is usually expressed in percent recovery.

Line 132: Fig2 can be omitted as all relevant information with regards to parallelism is given in the text (line 128-131)

Line 133-134: If sampling for SN was insufficient, the entire data set should be omitted from the analyses and subsequently the results section. For the description of the intervention-effect, please also indicate individual fold increases in uGC and fGC concentrations. Subsequently the discussion should be adjusted accordingly.

Line 143: When comparing signal lag time, the authors should try to justify the references used for comparison by outlining similarities in study setup and housing.

Line 157: Is there any evidence for an assumed faster MR for TT? Please also compare in terms of number of samples produced post-interference, even if not collected/used.

Line 159: Can’t follow the argument for a higher immunoreactivity, please explain in more detail.

Line 165: Difficult to understand, please rephrase. Maybe separate findings for urine and faeces. Second sentence is somewhat repetitive to the first statement (line 163) and thus should be omitted.

Line 238: Insufficient explanation, please specify physiological parameters shown including unit

·

Basic reporting

The language used and the structure of the manuscript is adequate and the figures are relevant. The cited references are relevant, but there are some more recent primate publications which could have been cited.

Experimental design

Ideally, the authors should have tested different EIA’s (specific for native cortisol, corticosterone and assays specific for its metabolites), especially for the fecal samples were cortisol is often heavily metabolized. Then the best assay could be selected (an important reference is Heistermann et al, 2006: Comparison of Different Enzymeimmunoassays for Assessment of Adrenocortical Activity in Primates Based on Fecal Analysis). The authors showed that the commercial essay for native cortisol can detect an extreme stressor (the handling and injection of saline), but it remains unclear if the assay can detect routinely experienced stressful events. If it is not possible to test additional assays, then the authors should test if the fecal cortisol assay used in this manuscript can also detect these routinely experienced stressors.

Validity of the findings

Although only samples from three animals were analyzed (fecal samples from two animals and urine from 3 animals), this is usually adequate to validate stress assays. But because only one assay was tested with a strong artificial stressor there is not enough significance to publish the results of this validation on its own.

Additional comments

The validation of stress assays is important and as the authors state, should be carried out for each studied species separately. If the application of this fecal cortisol essay is applied in a study of golden snub-nosed monkeys, then the validation results of this manuscript should be published in that study. For example, the authors could investigate the effect of tourist presence on the captive monkeys.

Some further comments:

Introduction
Line 48: Why it is necessary to establish a baseline for CC change? It is important that the applied assay reliably detects a stressor.

Methods
Line 67: Why is provisioning of the animals mentioned in the ethic statement?
Line 79: I suggest the authors mean: The acute imposed stress is stronger than the impacts of tourists and diurnal rhythm. It would be interesting to know what the impact on stress levels of these factors are.
Line 81: More information is needed on how the animals were handled. The injection was probably a minor stressor compared to capture.
Line 88: According to the graphs and the raw data samples were collected 4 days before and 4 days after the manipulation.

Results
Line 112: Is there an explanation why the urine was diluted so much more in males?
Line 133. The authors should state if fecal samples were analyses from SN or not. Was rain not a problem for urine sampling?

Discussion
Line 144. Is cortisol lag time in primates related to size? The other species mentioned here are larger than the snub-nosed monkeys.
Line 155: Is there any evidence that cortisol excretion lag time is correlated with gut transit time in primates?

---

## Round 0.2 · Major Revisions

Whereas both reviewers are happy with the overall content of the manuscript, they have major language issues. I agree with them that the manuscripts needs major revision by a native English speaker. Particularly the discussion is difficult to read. The reviewers have thoroughly reviewed the manuscript and made extensive suggestions for improvement. Please take each of these into account. However, this does not replace the need for a native English speaker's revision.
Please also start your discussion with a short summary of your findings: Our results show that ... before you start interpreting them. Avoid going into too much detail of results (omit mentioning the sample size repeatedly) to avoid repetition.

Reviewer 2 previously raised objection against the validity of your approach (at that time the comment was: "Although only samples from three animals were analyzed (fecal samples from two animals and urine from 3 animals), this is usually adequate to validate stress assays. But because only one assay was tested with a strong artificial stressor there is not enough significance to publish the results of this validation on its own."). This concern needs to be included into the discussion.

Reviewer 1 ·

Basic reporting

The revised MS by Chen and colleagues adds some valuable methodological information to the field of wildlife endocrinology by demonstrating the suitability of an EIA to monitor stress-related hormone values non-invasively in an endemic Asian colobine. However, the MS still needs some revision before it could be considered for publication.

Experimental design

no objections

Validity of the findings

no objections

Additional comments

The revised manuscript by Chen et al., describing the validation of an enzyme-immunoassay for determining urinary and faecal glucocorticoid metabolite concentrations in golden snub-nosed monkey, improved with most of the previous issues raised been clarified. However, there are still some aspects which need further clarification or additional information. In this regard, I again strongly advise to seek the assistance of a native speaker to improve the language quality of the MS.

The following indications for changes refer to the Word document with track changes

Title: I suggest ‘Assessing urinary and fecal metabolite concentrations as stress indicators in golden snub-nosed monkeys (Rhinopithecus roxellana)’

Running head: I suggest: ‘Non-invasive stress monitoring in R. roxellana’

Line 29: ‘…samples are a reliable source for determining iCC concentrations and to determine a suitable sampling regime.’
Line 34: ‘distinct rises in’
Line 48: ‘… the lag time is comparatively consistent for primates; for example it is …’
Line 53: ‘primate species’
Line 82: ‘..or fecal cortisol metabolite concentration (CC)…’
Line 83: ‘used as an indicator of perceived stress in …’
Line 86: ‘…validate an enzyme immunoassay for detecting urinary and fecal cortisol metabolites, and…’
Line 87: ‘…for captive golden snub-nosed monkeys.’
Line 91: ‘…several years ago and were brought to … as the conditions for reintroduction into the wild were not suitable at this stage.’
Line 108ff: I can’t see this as an argument for imposing a more substantial stressor, please omit and rephrase remaining sentence accordingly
Line 110: Please omit first half sentence
Line 112: Omit sentence starting with ‘So the capture…’
Line 114ff: Please rephrase indicating that interventions were conducted in serial order, last 10-15 min per individual.
Line 122: ‘procedure of urine and feces for 20 days prior the start of sample collection’
Line 125: ‘…opportunistically during daytime…’
Line 127ff: Please rephrase; Sample material has always been collected from the outside of the enclosure to avoid…’
Line 129: ‘Prior injection 4 urine and 3 fecal samples have been collected’ – Is that in total or on average per individual?
Line 131: Omit sentence
Line 133: ‘…after every collection’
Line 134: A minimum of 0.5ml of urine were collected by …’
Line 137: ‘… if it was in reach of the investigator’
Line 138: Omit sentence
Line 148: ‘…in range of the respective standard curve’
Line 179: ‘…(respective iCC concentrations were determined…’
Line 182: Please refer to respective figure
Line 204: Please omit first half sentence
Line 208: ‘The number of fecal samples for SN…’
Line 209: ‘…, so we excluded this individual from all subsequent analyses’
Line 212: ‘… CC concentrations increase about … above the individual baseline level. The individual mean fecal CC concentration of TT…’
Line 214: ‘…fecal CC concentrations…’
Line 223-226: Suggest swop sentences
Line 232: 08:-28 ?
Line 233: ‘… voided at night’
Line 234: ‘… The increase in hormone concentration detected post-intervention also supports this explanation’
Line 258ff: Please add a brief paragraph discussing the individual variation in baseline urinary and fecal iCC concentrations, especially in relation to the sequence the individuals were handled
Legend Fig 2: Please omit ‘of EIAs’
Legend Fig 3: ‘Longitudinal profile of urinary and fecal iCC concentrations for …’
Table 1: It would be beneficial if the authors could add information regarding the time samples could not be collected.
Fig 3: Please always use either dashed or solid line for the fecal hormone profiles

·

Basic reporting

Basically the same comments as in the first review. The authors have only addressed the general comments.

The language quality should be improved by a native speaking colleague, I have only made some suggestion below.

Experimental design

More information has been provided on fecal and urine collection, see additional comment below.

Validity of the findings

Same as in previous review

Additional comments

Change Title: Assessing the utility of urinary and fecal cortisol as an indicator of stress in golden ….
Also change the running head

Abstract
Line 22: Delete the added immunoreactive cortisol metabolites, this is mentioned in lines 23-24
Line 25: …whether CC can be detected in urine and feces to reliably assess stress…
Line 29: …was detected with a commercial EIA kit and showed the expected sharp rises..
Line 33: … is within one day of a potential stressful event

Introduction:
Line 37: Be more precise. Especially in feces, there is rarely a large amount of native cortisol present. And also describe the advantages over blood samples.
Line 46 … which are effected…
Line 46: Although not always the case… (reference is needed for this).
Line 50: Be more precise here, and discuss native cortisol and metabolites of cortisol in feces and urine.
Line 57-60: Do not introduce abbreviations if they are not used again in the manuscript.
Line 65. Use monitoring instead of safe guarding
Line 69: But you only validated one immunoassay kit!

Methods:
Line 73-74: Rephrase these sentences and move to animals and housing.
Line 83: Did they use this cage to sleep and rest in?
Line 98: What does opportunistically mean here?
Line 100. The investigators did not enter the cages to avoid …
Line 103: Was the entire fecal bolus collected, or a portion? Ideally, the feces should be mixed before a portion is collected.

Results:
Line 165 -167. Except for a few samples … No need to repeat the numbers if they are in the table. But there are quite a few samples uncollected (does this include the discard samples?). You need to state why this is mentioned here and in the table, and why they are included in the total sample (the calculate excretion rates?). does this include feces voided during the night? Maybe explain this in the method section.
Line 170: It is still unclear why urine collection worked for CN, but fecal sampling failed.

Discussion:
Line 181: First discuss the more generalized outcomes, before discussing differences between individuals.
182: Provide rates rather than refer to table 1.
191: …voided during the night. How often? Was this recorded? Rephrase the next sentence.
Line 193: The peak value represents a 7 fold increase compared to bas3eline levels, etc.. Is it correct that you suggest that that a higher peak at a later time would is expected for TT?
Line 195. This sentence is not clear. What do you mean with: skew to 23 hours?
Line 199: Can you confirm that the animals did not have diarrhea after handling?
The last paragraph is difficult to comprehend and needs to be rewritten.

Fig.2: Use the same line style for all individuals. Use the same scale for each individual (urinary cortisol)

---

## Round 0.3 · Minor Revisions

Let me apologize for the delay in providing you with a response to your latest submission to PeerJ. Unfortunately, the editor handling your article is no longer available and so I was asked to take over in their place. I have read your most current revision of your article, the two reviewers’ comments, as well as the previous submissions of your article and the associated reviewers’ comments.

Considering your latest submission, both reviewers note that you have taken considerable efforts to address their previous concerns and to improve the clarity of your article. I agree with them. The reviewers have also provided detailed feedback as to how you might further enhance your article. Indeed, Reviewer 1 has even taken the time to provide thorough edits to your manuscript, which you can see in the annotated article attached, as well as some questions therein that you must address. Below, I provide some additional comments of my own, but I believe that if you are able to respond to these, and also to the comments provided by the two reviewers, your article will likely be suitable for publication in PeerJ.

My general comments:

You used capture and injection for the acute stressor. How often is this procedure conducted at the facility where the monkeys were housed? Have they experienced this before? If so, how frequently and how recently? Please provide this information in your article.

The final paragraph of your Discussion is pure conjecture. Please remove it and focus on the validation of the methods, rather than the interpretation of them if you do not have appropriate (e.g., behavioral) data to support your claims. Additionally, I think it would be worthwhile if you were able to provide discussion of the application of your (validated) methods. For example, you mention that it is possible that the monkeys you studied might have shown signs of stress in response to tourists. Would this be something that you could test in the future? How might your methods be applied at other institutions or in the wild? How could your methods benefit this species more generally, especially considering its IUCN Endangered status.

Comments on specific phrasing within your manuscript:
Line 106 – “injection, subjects fully recovered in minutes” – this should be rephrased as “injection, subjects appeared to recover in minutes”

Line 115 – “and they usually rested and defecated near the edges of the cage, resulting in entering cages was unnecessary for sample collection” would be more clearly worded thus “the monkeys usually rested and defecated near the edges of the cage, meaning entering cages was unnecessary for sample collection.”

Line 203 – “Based three captive adult of a high-stress situation (capture and injection), our results show that it is reliably to detect immunoreactive cortisol (or metabolites) in the urine and feces of golden snub-nosed monkeys by EIA” would be more clearly worded thus “Based on the responses of three captive adult golden snub-nosed monkeys in a high-stress situation (capture and injection), our results show that it is reliable to detect immunoreactive cortisol (or metabolites) in the urine and feces of this species by enzyme immunoassay (EIA)”

Line 207 – “But because only one assay was tested with a strong artificial stressor, in the application we should concern its significance when using other assay.” I do not understand this sentence. Please rephrase it.

Line 216 and elsewhere, where you refer to the monkeys with the acronym labels, I suggest explicitly noting that you are referring to monkeys to avoid confusion with the other acronyms (e.g., CC and EIA) that you use in your paper. E.g., where you state “First, TT had urinary and fecal baseline levels more than twice those of QQ, and urinary baseline cortisol levels nearly three times higher than SN.” You might wish to write “First, male monkey TT had urinary and fecal baseline levels more than twice those of male monkey QQ, and urinary baseline cortisol levels nearly three times higher than the female monkey SN.”

Please avoid retractions throughout e.g., line 223 “so we hadn’t collected it.” should be “so we had not collected it.”

Reviewer 1 ·

Basic reporting

As mentioned before, the language quality of the MS needs to be improved.

Experimental design

n/a

Validity of the findings

n/a

Additional comments

The revised version of the manuscript by Chen et al., describing the validation of an enzyme-immunoassay for determining urinary and faecal glucocorticoid metabolite concentrations in golden snub-nosed monkey, improved, but still needs further revision and additional information. As mentioned before, the language quality of the MS needs to be improved.

I indicated possible alterations with track changes and comments in the word document attached.

Annotated reviews are not available for download in order to protect the identity of reviewers who chose to remain anonymous.

·

Basic reporting

English has improved. Suggestions for further improvements are below.

Experimental design

Still missing in the results is an analysis of excretion frequency. Especially since the authors refer to this in the discussion. Could be shortened.

Validity of the findings

Considering the low sample size the discussion is too speculative.

Additional comments

Abstract
Line 23-26. Rewrite the two sentences. First sentence is incomprehensible. In the second sentence emphasis the importance of validating the essay that measures CC in feces and urine

Introduction
Line 38: Mention and Cite Add to dictionary work on baboons as an example for this.
Line 40: … to assess the stress response … (also I other places throughout the manuscript)
Line 42: Change to: Although cortisol is the major glucocorticoid in most primate species.
Line 48: Also state here that essays specific for a range of metabolites and native cortisol exist.
Line 54: … for the appearance of cortisol in feces …
Line 60: Change to: Thus, for each species, it also important to identify the lag time for cortisol excretion (or its metabolites) besides validating the method to monitor the stress response.
Line 74: CC cannot stand for everything (fecal cortisol, metabolite concentration etc.). Rethink the use of the abbreviation CC, maybe just use the term cortisol throughout the manuscript.
Line 76 Change sentence: In this experiment, by handling three golden snub-nosed monkeys and injecting a saline solution we stimulated a potential stress response.
Line 78: You still aim to measure metabolites, although the essays are specific for cortisol?

Methods
Line 84: Delete the second sentence.
Line 97: change title to Validation Experiment
Line 100. … potentially acute stressor ….
Line 115: replace …resulting … with … thus, entering …
Line 116: It is better to present the average and range of samples collected here.
Line 134. Wasser et al (2000) and Fan et al. (2013), and the instructions provided by the EIA kit manufacturer.
Line 151: Usually internet addresses can be cited as references and can be included in the reference list. Check with the journal.
Line 158: Regression curves of measured and expected cortisol concentrations are presented in Figure 2.

Results:
Line 186: Is suggest that the authors explain why fecal sample size was low for SN, and how many were collected. Either in the results or in the methods.

197: Replace sentence with : Lag time of urinary and fecal cortisol excretion was examined relative to the timing of injections.

Discussion
Line 203: Based on samples from three …
Line 204: Why metabolites?? Delete.
Line 205: …with a commercially available EIA specific for cortisol.
Line 207. Is there a reference for the “usually adequate”? Next sentence is confusing. Are the authors planning to use another assay?
Line 209: … lag time for this study …
Line 212: Don’t start a new paragraph. Be clear here that values are only from tow animals (5 and 23 hours, not between).
Line 219: Not clear why TT should have higher peak CC later. Enough samples seem to have been collected. Too much speculation in this paragraph.
Line 236. But this was expected considering the acute stressor. Again, there seems to be too much speculation considering that data is only available from 3 animals. The authors should rather end the discussion addressing the aims of the study.

---

## Round 0.4 · Minor Revisions

Thank you very much for making considerable changes to your manuscript in response to my and the reviewers' comments. I believe that you have addressed all of my, and the reviewers', previously-noted concerns, and given that, and the extensive improvements you have made to your manuscript, I have decided not to send your manuscript out for review again.

However, I did notice a few typos and a couple sentence that were unclear within this latest version of your article. I have highlighted them and provided suggested edits in the annotated pdf file attached. If you can attend to these minor edits and resubmit your manuscript, I will be happy to accept your article for publication in PeerJ.

---

## Round 0.5 · accepted · Accept

Thank you for making those final few edits to your manuscript that I requested. I am pleased to inform you that I have accepted your article for publication in PeerJ.